# Perceived impact of information signals on opinions about gluten-free diets

**Franklin Bailey Norwood** *

Department of Agricultural Economics, Oklahoma State University, Stillwater, Oklahoma, United States of America

* bailey.norwood@okstate.edu

## Abstract

Understanding how people assimilate different types of information for food choices is integral to improving knowledge about diet and human health. This study evaluates the impact that 10 information signals have on the perceived healthiness of gluten. Signals include non-social signals such as personal eating experiences, scientific studies, and advice from doctors, but also includes social signals such as recommendations from attractive people, social media, the layout of a grocery store, and celebrities. An online survey of over 1,000 Americans is administered using indirect questioning where subjects are presented with a hypothetical other person and asked how the various signals would impact that person's opinion of gluten-free diets. Results show that advice from an attractive person is thought to have a slightly larger impact than reading about a new study regarding gluten, and seeing a grocery store develop a new gluten-free section has a larger impact than learning a celebrity consumes a gluten-free diet.

**Data Availability Statement:** Data has been submitted to DRYAD and is available at https://datadryad.org/stash/dataset/doi:10.5061/dryad.05qfttf12.

**Funding:** Funding was made possible by the Barry Pollard MD / P&K Equipment Professorship in

## Introduction

It is increasingly clear that consumers rely on more than scientific publications and medical advice to make decisions about food and health. For example, the growing popularity of gluten-free foods is hard to explain solely based on recent scientific findings or new paradigms in medical research. True, there was initially some evidence suggesting the existence of gluten-sensitivity among the non-Celiac population [1]. However, more recent evidence is mixed [2–5] and there is still no consensus on whether gluten sensitivity is a medical condition [6, 7]. While there is a protocol for diagnosing gluten sensitivity [8], people may self-diagnose or infer that gluten is unhealthy in ways other than irritable bowels. If gluten sensitivity exists, it is a condition in which little is understood, and affects only a small portion of the population. Certainly, the medical profession does not consider foods containing gluten to be less healthy for the general population than their gluten-free counterparts [7].

Yet, about 15% of people think they may be sensitive to gluten, and one in five think health can be improved by keeping gluten off their plates, as shown in Fig 1 below [9]. The ubiquity of 'gluten-free' food labels, even on foods that never contain gluten, can be seen in any grocery

Agribusiness. The funders had no role in study design, data collection and analysis, decision to publish, or preparation of the manuscript.

**Competing interests:** The authors have declared that no competing interests exist.

store, attesting to industry reports [10] that the market for gluten-free foods has risen and will continue to rise.

For a condition that is supposed to be medical in nature, gluten sensitivity has a rather strange social aspect to it, at least from popular media. It is common for politicians, movies, and television to depict gluten sensitivity as existing primarily among Americans on the political-left, yet recent research suggests it is a popular eating behavior among Trump supporters as well [9]. Survey evidence even suggests that some people consider a gluten-free diet to be a successful weight-loss strategy [11].

It only takes a moment of reflection to acknowledge that much of our beliefs about health and food do not stem from carefully vetted research and advice from medical institutions, although they obviously play an important role. Conceptual models of food choices constructed from interviews document social factors as playing an important role, and empirical studies support this claim [12, 13].

Social information signals concerning gluten's impact on health includes personal testimonies. Such testimonies may come from friends, people we admire such as celebrities, or people who seem to take good care of their bodies. These are signals provided by specific people we encounter in our daily life. Other social signals emerge from the collective actions of many people, most of whom we do not know. Consider the layout of a grocery store. The more other people attempt to remove gluten from their diets, the more grocery stores will market towards these individuals, perhaps creating new sections catering solely to gluten-free products. This could just be stores attempting to profit from health fads, but it could also signal the fact that many other people are learning that gluten is bad for their health. Or consider the fact that January 13 is the official gluten-free day. True, there is a 'day' for all types of trivial matters. On July 14, you can celebrate Bastille Day, commemorating an historically important day for

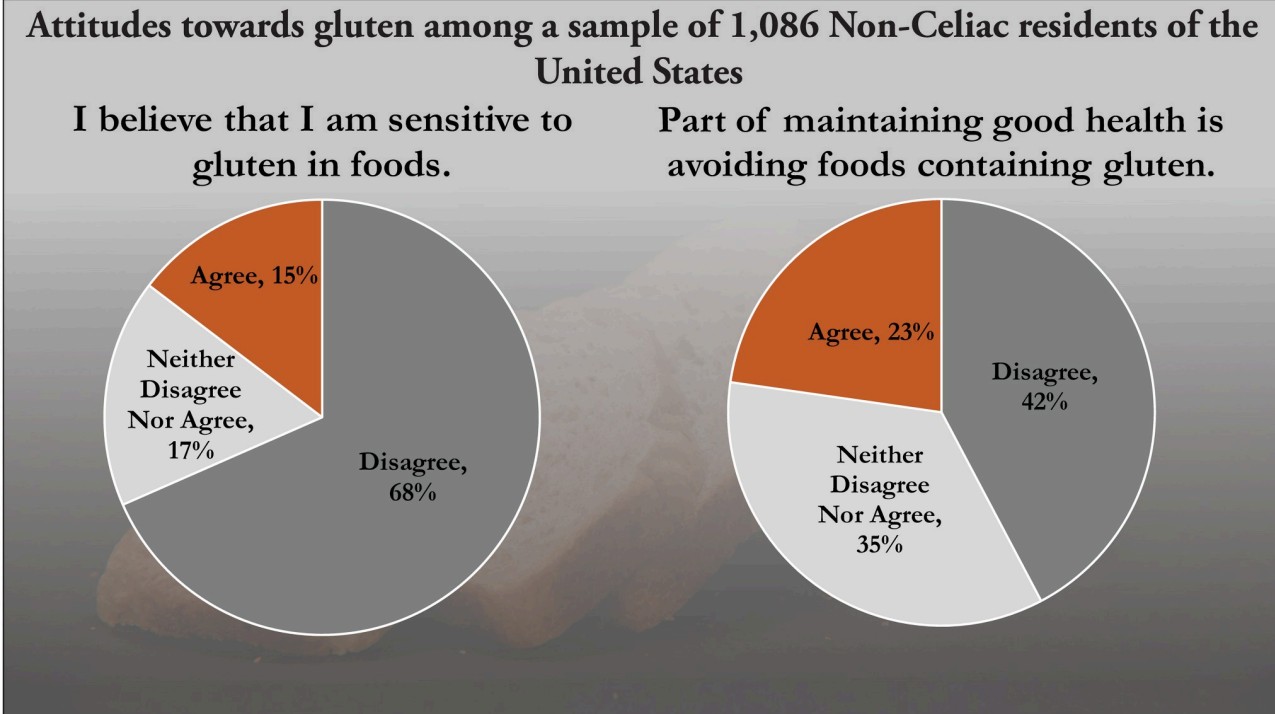

**Fig 1. Attitudes towards gluten as held by a representative sample of U.S. citizens.** Source: original analysis of data reported in [9].

France. On that same day you can also celebrate National Nude Day. However silly National Nude Day may be, the fact that it exists says something about people, and the fact that there is a gluten-free day might suggest the anti-gluten movement is real and, thus, gluten may be bad for health.

What type of social information signals have the largest impact on opinions about gluten-free diets, and how does their impact compare to other signals such as a personal eating experience or a doctor's recommendation? This is an exploratory analysis of how a variety of potential social signals may impact people's opinions of gluten-free diets, meaning we did not design the survey experiment to test any explicit hypothesis. Rather, we sought to explore how a number of plausible signals about food and health might be interpreted. An internet survey was conducted where respondents were presented with a variety of signals and asked to predict the impact the signal would have on a hypothetical other person's opinion of gluten-free foods. For comparison, in addition to seven social signals, three non-social signals are included: doctor recommendation, a personal eating experience, and reading about a new study concerning gluten. The next section describes the subjects who took the survey and the survey instrument. It is followed by a section on the statistical procedures employed. Then a section on the results is provided, which is followed by a description of the study's limitations, and then a general discussion.

## Materials and methods

The design and implementation of the study was evaluated and approved by the Institutional Review Board at Oklahoma State University, application AG-19-36. Explicit consent was not obtained, as consent to participate was implied if the participant chose to take the survey. Below we describe the participants recruited and the survey design.

### Participants

An internet survey of Americans was conducted in the fall of 2019 using a sample acquired by the Qualtrics company, which uses an opt-in panel of respondents that ensures a representative sample of Americans in terms of key demographics like gender and ethnicity. These are respondents who are recruited to take online surveys in exchange for compensation such as gift cards, airline miles, and the like. The original sample contained 1,535 respondents. After removing those with incomplete responses, a total of 1,317 respondents remained. Descriptive statistics of the sample are shown in Table 1, as well as their counterpart statistic for the U.S. population as determined by the 2010 Census and the American Community Survey.

A number of the demographic variables closely resemble the U.S. population and suggest a representative sample. For example, the sample is 50.93% female, which closely matches the U.S. population of 50.8% female. However, other sample demographics depart from the population, like the 54% of respondents 25 years or older in the sample with a bachelor's degree, compared to 32% for the United States. Moreover, research has shown that samples from opt-in panels can differ from the population in a number of other features, such as the percent of households with an unemployed member looking for work [14]. Table 1 shows that 23% of the households had at least one unemployed member where, according to the American Community Survey, the percentage for a representative sample of the population during normal economic years is around 8%.

To correct for differences in the sample relative to the population, a sample balancing algorithm used by [15] is employed to calculate weights for each respondent, such that each weighted statistic for the sample in Table 1 equals the statistic for the U.S. population. For

**Table 1. Descriptive statistics of 1,317 survey respondents.**

| Variable | Representation in Sample | Representation in US Population[a] |
|---|---|---|
| Female | 50.93% | 50.80% |
| Male | 49.07% | 49.20% |
| ≤ 34 years of age | 30.21% | 27.38% |
| 34 < years of age ≤ 54 | 37.14% | 33.67% |
| 54 < years of age | 32.64% | 38.95% |
| White ethnicity only | 74.71% | 76.30% |
| Black ethnicity only | 14.43% | 13.40% |
| Other ethnicity | 10.96% | 10.30% |
| Hispanic | 17.43% | 15.30% |
| Household income ≤ $35,000 | 28.07% | 27.89% |
| $35,000 < Household income ≤ $75,000 | 32.36% | 29.21% |
| $75,000 < Household income | 39.57% | 42.91% |
| Northeast region | 18.36% | 17.10% |
| Midwest region | 19.07% | 20.80% |
| South region | 42.57% | 38.30% |
| West region | 20.00% | 23.90% |
| Household has at least one unemployed member looking for work | 22.64% | 8.30% |
| Households with three or more members | 50.57% | 62.88% |
| Respondents 25 years or older with a bachelors degree | 54.00% | 31.50% |

[a] Statistics for U.S. population demographics are acquired from the U.S. Census Bureau, using statistics from the 2010 Census and the American Community Survey for years 2014–2018.

example, using these sample-balancing weights, the weighted proportion of households with three or more members equals 62.88%, equaling that of the population.

## Materials

To measure the perceived impact various information signals have on people's opinions of gluten-free diets, the survey presented respondents with a number of these signals and hypothetical other people. Subjects were then asked to indicate the impact each signal will have on the other person's attitude towards gluten-free diets. This survey is presented in S1 Appendix. The first part of the survey concerned the demographic questions shown in Table 1. The second part contained 10 to 14 questions regarding opinions on voting, honesty, and foods that are not related to this study and are used for a separate analysis. The third part of the survey then concerned the relationship between information signals and opinions on gluten.

This third part begins by asking respondents if have heard of gluten, whether they can identify foods with gluten, and their general opinion regarding gluten. Only 4.10% of the sample said they had never heard of "gluten-free" foods, and while they are retained in the sample, a sensitivity analysis will show their inclusion has only minor impacts on the empirical results. When given a list of five foods (bread, meat, honey, tomatoes, and lettuce), and asked which are most likely to contain gluten, 82.71% of the sample correctly identified bread. When asked to choose one of four options best reflecting their opinion about gluten, 8.71% said it was unhealthy for everyone, 51.64% said it was unhealthy for some people, 11% said it was healthy for everyone, and 28.65% were unsure.

**Signals.** Subjects were then told they would be presented with 10 questions with each one containing a picture of a person and an event (referred to here as information signals) where

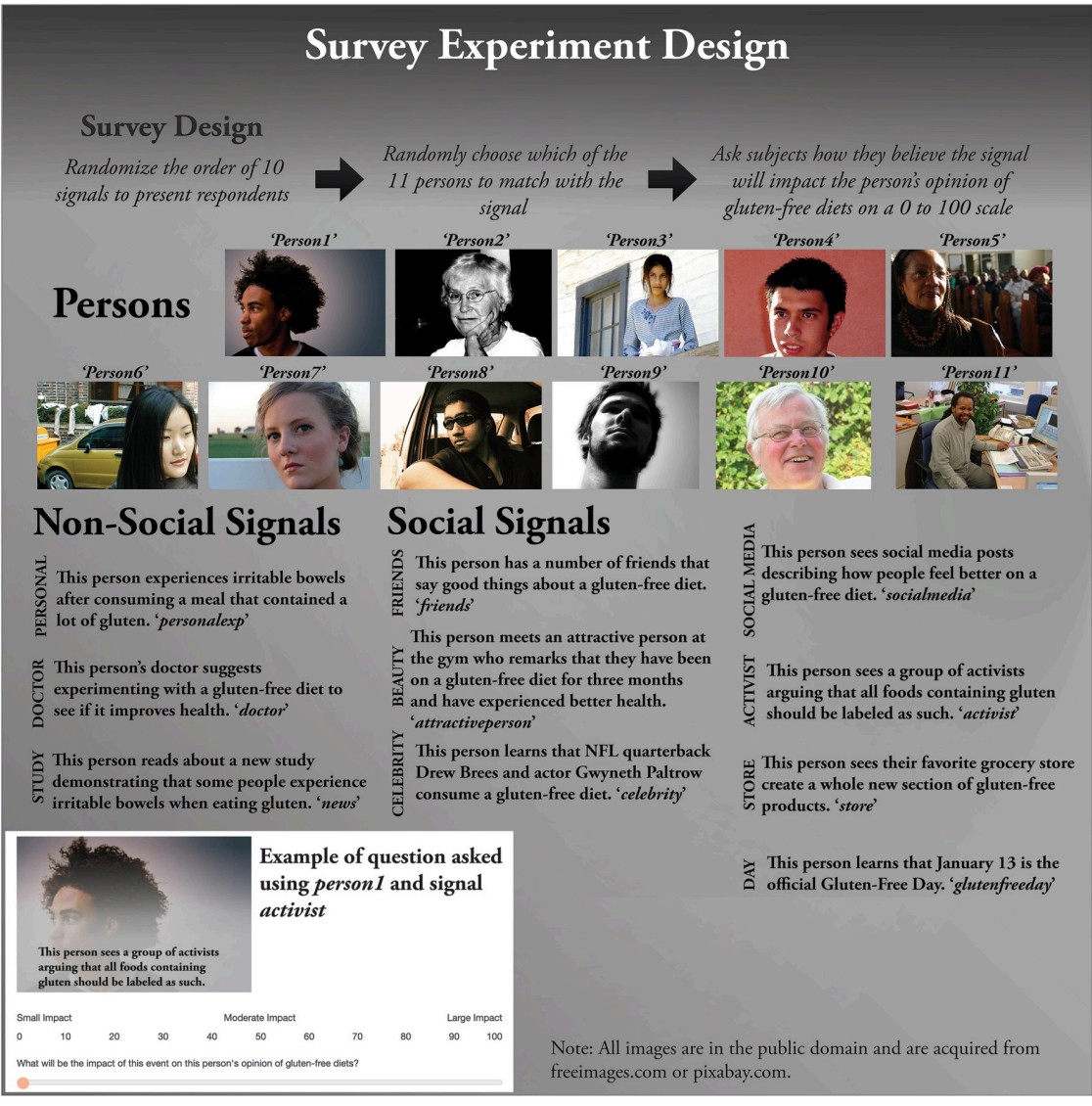

**Fig 2. Signals, persons, and experimental design of the survey.**

the person is exposed to information regarding gluten-free diets. For each question subjects were told they would be asked how large of an impact they believe the event will have on the person's opinion of a gluten-free diet. The type of signals, persons, and the question format is illustrated in Fig 2 below.

The signals chosen in Fig 2 were selected to represent both non-social and social information signals. The non-social signals were designed to reflect either personal experiences or objective sources of information. One non-social signal is personal experience (labeled *personexp* in Fig 2), described as experiencing irritable bowels after consuming a meal containing gluten. This is a documented reason for adopting a gluten-free diet [16]. Notice this refers to a single meal, so it represents a single personal experience and not necessarily a series of bad experiences with gluten. Objective sources of information can come from doctor recommendations or scientific studies. The *doctor* signal says that the person's doctor recommends experimenting with a gluten-free diet to see if it improves health. This is not the same as a

doctor saying gluten-free diets are generally healthier, only that it might be for some people. The signal referring to scientific studies (*news*) was described as a person reading about a new study demonstrating how some people experience irritable bowels from consuming gluten. As with the *doctor* signal, the *news* signal only says that a study shows some people might be gluten-sensitive, not that gluten is unhealthy for everyone.

The social signals are intended to represent the beliefs of certain social groups that might praise the health effects of a gluten-free diet. The signals *friends*, *attractiveperson*, and *celebrity* refer to influences from specific people. *Friends* refers to an instance where a person "has a number of friends that say good things about the gluten-free diet." It makes sense that people who are physically attractive take good care of their bodies, including eating a healthy meal. The *attractiveperson* signal refers to a scenario where an attractive person met at the gym says they feel better on a gluten-free diet. Given that celebrities are thought to possess great influence, the *celebrity* signal refers to learning that NFL quarterback Drew Brees and actor Gwyneth Paltrow consume a gluten-free diet. These two celebrities were chosen because at least one of them should be appealing to most subjects.

The other social signals refer not to specific individuals but the collective actions of others who are likely strangers. A person might be influenced by seeing social media posts describing how people feel better on a gluten-free diet, and that is how the *socialmedia* signal is described. A person seeing activists argue that all foods containing gluten should be labeled as such might believe the activism is based on real health problems, so the *activism* signal is included as well. Walking into one's grocery store and seeing a new display devoted solely towards gluten-free products might be viewed by some as a marketing gimmick, but others may infer from it that many others are benefitting from a gluten-free diet, so this signal (*store*) is included. Finally, the fact that January 13 is the official gluten-free day suggests that the diet is quite popular with many, so the signal *glutenfreeday* is included as well.

**Hypothetical other persons.** Although the goal is to measure how each of the signals would impact people's opinions about a gluten-free diet, we deliberately avoid asking the subject how it would impact their own opinion. Instead, we ask how it would impact the opinion of a hypothetical "other" person. This is done to avoid social desirability bias, whereby a person misrepresents their true attitudes in order to appear more desirable to others. Research has shown that when you ask a person about the behavior of other people you often obtain better information about their own behavior, compared to when you directly ask them about their own behavior. For example, if you want to know whether a person would vote for a Muslim presidential candidate, you might obtain a more truthful response if you ask whether the average American would vote for one rather than if you ask if they, themselves, would. In regards to gluten, a person might be embarrassed to admit that they allow social media to influence their views, so even if social media has a larger impact on their views of gluten-free diets than their doctor's advice, they would be reticent to admit so in a survey. Respondents would not experience the same hesitation to say so of others, however, so the indirect questioning is used to provide a more realistic depiction of the subjects' true beliefs than if they were asked directly [17–19].

Indirect questioning often asks subjects to predict the behavior of the "average American," but it was suspected that doing so would cause the respondent to imagine a white female, as that seems to be the demographic most commonly associated with such diets in popular media. It thus seemed important to give the subject a picture of an individual to consider in the question. Then, so that the survey results as a whole do not pertain to pictures of any one particular demographic, the type of persons presented in the survey varied randomly across signals and surveys. Fig 2 shows the 11 types of persons used in the survey.

Of these persons, three are African American, four are white, two are Hispanic, one is Asian, and one is a more ambiguous ethnicity. They reflect a wide array of ages, from young to middle aged to elderly. By randomly varying the person associated with each signal, the survey ensures that the average response is not reflective of any one particular demographic; instead is a mix of the 11 persons. Subjects are asked about all 10 signals, but the order in which each signal appears in the survey is randomized.

For each signal/person combination presented, the subject is asked, "What will be the impact of this event on this person's opinion of gluten-free diets?" The impact is measured on a 0 to 100 scale where a larger value refers to a larger projected impact. Although the impact should largely be in favor of gluten-free diets, the question does not require this; it only asks about the size of the impact, not whether it causes the person to look upon gluten-free diets more or less favorably.

## Statistical procedures

### Averages and sign tests

A number of different empirical procedures are used to analyze the survey results. The first includes a simple average of the impact scores. A simple average is an unbiased estimator of each signal's importance because both the question order and the person matched with each signal is randomized. Additionally, a weighted average is performed using the weights calculated from the sample balancing algorithm to better represent the views of the U.S. population and not just the sample.

One problem with comparing average scores across signals is that different subjects may use different mental scales when answering the survey questions, making it difficult to compare a score from one person to the score of another. For example, one respondent may consider an impact score of 60 to have more or less actual impact than another person who also provides a score of 60. However, if the percentage of subjects who provide a higher score to signal A than signal B is statistically greater than 50%, one can say signal A is indeed thought to have a larger impact. As such, nonparametric sign tests, as described by a 2015 study [20], are used to determine if one signal is consistently assigned a larger impact than another signal across individuals. The null hypothesis of this test is that the median of the difference between impact scores of two signals is zero, and makes no assumption about the statistical distributions of the scores.

### Regression 1

The signal impacts are also analyzed using regression analysis. The first regression models the projected impact as conditional on the signal being evaluated, the demographic profile of the respondent, and the hypothetical other person being matched with the signals. This regression is stated as follows.

$$
\begin{aligned}
Impact_{ij} = {} & \beta_1(activist_{ij}) + \beta_2(celebrity_{ij}) + \beta_3(glutenfreeday_{ij}) + \beta_4(doctor_{ij}) + \\
& \beta_5(friends_{ij}) + \beta_6(news_{ij}) + \beta_7(personalexp_{ij}) + \beta_8(attractiveperson_{ij}) + \\
& \beta_9(socialmedia_{ij}) + \beta_{10}(store_{ij}) + \alpha Z_{ij} + \tau P_{ij} + e_{ij}
\end{aligned}
\tag{1}
$$

In (1), $Impact_{ij}$ is the projected impact of the signal for the ith person on the jth question, where $0 \leq Impact_{ij} \leq 100$. The variables $activist_{ij} \ldots store_{ij}$ are indicator variables for the type of signal, and their description was given previously in Fig 2. For example, if the signal concerns the person seeing "activists arguing that all foods containing gluten should be labeled as such" then $activist_{ij} = 1$; otherwise $activist_{ij} = 0$. The vector $Z_{ij}$ refers to indicator variables for

the following demographics: (1) one indicator variable for female; (2) one variable for under 35 years of age and one variable for over 55; (3) one variable for white and one variable for black ethnicity; (4) one variable for Hispanic; (5) one variable for pretax annual household income below $30,000 and one for above $75,000; (6) one variable for northeast region, one for midwest, and one for south; (7) one variable for households with three or more members; and (8) one variable for respondents 25 years of age or older with a bachelor's degree. As such, the default demographic is a male, without a bachelor's degree, 35 to 54 years old, other ethnicity, non-Hispanic, in the western United States, making between $30,000 and $75,000 of household income, and in a household with less than three members.

The vector $P_{ij}$ refers to the hypothetical other person randomly chosen for the ith person in the jth question. Although these variables are not necessarily needed as they are randomly matched with signals in the survey, including them provides information on what types of hypothetical other people are thought to be more impacted by signals. This vector takes the form $P_{ij} = [Person2_{ij} \ldots Person11_{ij}]$, where the identity of *Person1* through *Person11* is given in Fig 2, and *Person2*$_{ij}$ is an indicator variable that equals 1 if *Person2* is matched with the signal and zero otherwise. As *Person1* is not included in the vector, it is the default category. No intercept is included in the model so that all signal indicator variables $activist_{ij} \ldots store_{ij}$ can be included. This implies that the coefficient $\beta_k$ should be interpreted as the average impact score for the kth signal provided by a person with the default demographic profile described previously and the default picture of *Person1*.

Each person answers 10 questions and, thus, the stochastic error $e_{ij}$ should be correlated across respondents: $e_{ij} \sim iid\, N(0, \delta_i^2)$, meaning the errors are identically and independently distributed across questions for the same person, but the errors for each person have their own fixed variance $\delta_i^2$. The model is estimated in STATA using ordinary least squares and the Huber and White robust estimate of variance [21, 22]. Because this model uses demographic variables as explanatory variables, sample balancing weights are not used in the estimation.

## Regression 2

The second regression is the same as Regression 1 except that the demographic variables in $Z_{ij}$ are omitted and a weighted regression is used to correct for differences in the sample and population demographics. The sample-balancing algorithm mentioned previously assigns a weight $W_i$ to each respondent, where the weights are calibrated such that weighted averages of the sample demographics conform to the actual average of the population demographics shown in Table 1. The ordinary least squares estimation is similar to Regression 1 except that, instead of minimizing the sum of squared residuals $\Sigma_i\Sigma_j(e_{ij})^2$, it minimizes the weighted sum of squared residuals $\Sigma_i W_i\Sigma_j(e_{ij})^2$. As with Regression 1, the Huber and White robust estimate of variance [21, 22] is used to account for the panel nature of the data.

## Sensitivity analysis

Three additional analyses are performed to evaluate the robustness of the results. It was previously mentioned that some respondents are less familiar with gluten than others, but the statistical analysis did not exclude anyone based on their knowledge of gluten. To investigate the impact of this decision on the results, Regression 2 is repeated twice with modifications. First, Regression 2 is estimated excluding those who say they have never heard of gluten-free foods. Second, Regression 2 is estimated excluding those who do not correctly identify which of the following foods contain gluten: bread, meat, honey, tomatoes or lettuce.

The empirical methods described previously use one number to describe the estimated impact of each signal for all respondents; however, there might be heterogeneity in the

responses across respondents. As such, while one signal may seem to have a larger impact than another for the sample as a whole, this comparison may be reversed for a subset of the sample. To allow for preference heterogeneity, a latent class model is estimated. This involves estimating a version of Regressions 1 and 2, but using only the signal explanatory variables. All respondents are assumed to belong to one of two different classes, denoted $c$, where each class has its own unique signal coefficients $\beta_{j,c}$. The error term is assumed normally distributed with a constant variance for each class of subjects and signal: $e_{i,j,c} \sim iid\, N(0, \sigma_{j,c}^2)$. Estimates are acquired using maximum likelihood, and $\tau_c$ is the probability of subject $i$ belonging to class $c$, which has to be estimated along with the other coefficients subject to the constraint $\tau_1 + \tau_2 = 1$.

$$
\begin{aligned}
Impact_{ij} = \Sigma_{c=1}^{2}\tau_c(&\beta_{1,c}(activist_{ij}) + \beta_{2,c}(celebrity_{ij}) + \beta_{3,c}(glutenfreeday_{ij})+\\
&\beta_{4,c}(doctor_{ij}) + \beta_{5,c}(friends_{ij}) + \beta_{6,c}(news_{ij}) + \beta_{7,c}(personalexp_{ij})+\\
&\beta_{8,c}(attractiveperson_{ij}) + \beta_{9,c}(socialmedia_{ij}) + \beta_{10,c}(store_{ij}) + e_{i,j,c}
\end{aligned}
\tag{2}
$$

## Results

### Averages and sign tests: Results

Although 1,537 respondents took the survey, only the results of 1,317 are analyzed. Observations are discarded if the subject does not answer all the survey questions needed for the statistical analyses. The average impact score across all 1,317 respondents for each of the 10 signals is shown in Fig 3. A larger bar indicates a larger predicted impact. These are simple averages without sample balancing, but results for weighted averages using sample balancing weights provides virtually identical answers and, thus, are not shown. The figure also shows the results of a two-sided sign test, where any two bars with the same letter are not statistically different, meaning the null hypothesis that they possess the same median cannot be rejected at the 5% level.

Not surprisingly, an unpleasant personal experience consuming gluten and a doctor's recommendation are thought to have the largest impacts on opinions about gluten-free diets. The average impact score for these two signals are in the upper 60s, meaning, on average, they are thought to have an impact between "medium" and "large." Sign tests also indicate the average scores for these two signals are statistically different, so an unpleasant experience consuming gluten is thought to indeed have the greatest impact on opinions on a gluten-free diet of all the signals considered.

Hearing a friend or an attractive person say good things about a gluten-free diet has the third largest impact, and the average scores from these two signals are not statistically different. Both are given statistically greater scores than reading about a new study saying gluten can cause irritable bowels. Social media has the sixth largest score and is followed by a grocery store creating a new gluten-free section. The influence of activists and celebrities are not statistically different and have the least impact of any signal except for learning that January 13 is the official gluten-free day. Note that while the sign tests do detect statistical differences between most signals, the average ratings all reside in the upper- to mid-moderate range. Even the least impactful signal is thought to have a moderate impact.

### Regression 1: Results

Table 2 shows the regression results from the model in (1) where impact scores are explained by the type of information signal, demographic profile of the respondent, and the hypothetical other person used. First consider the coefficients for the information signals *activist . . . store*.

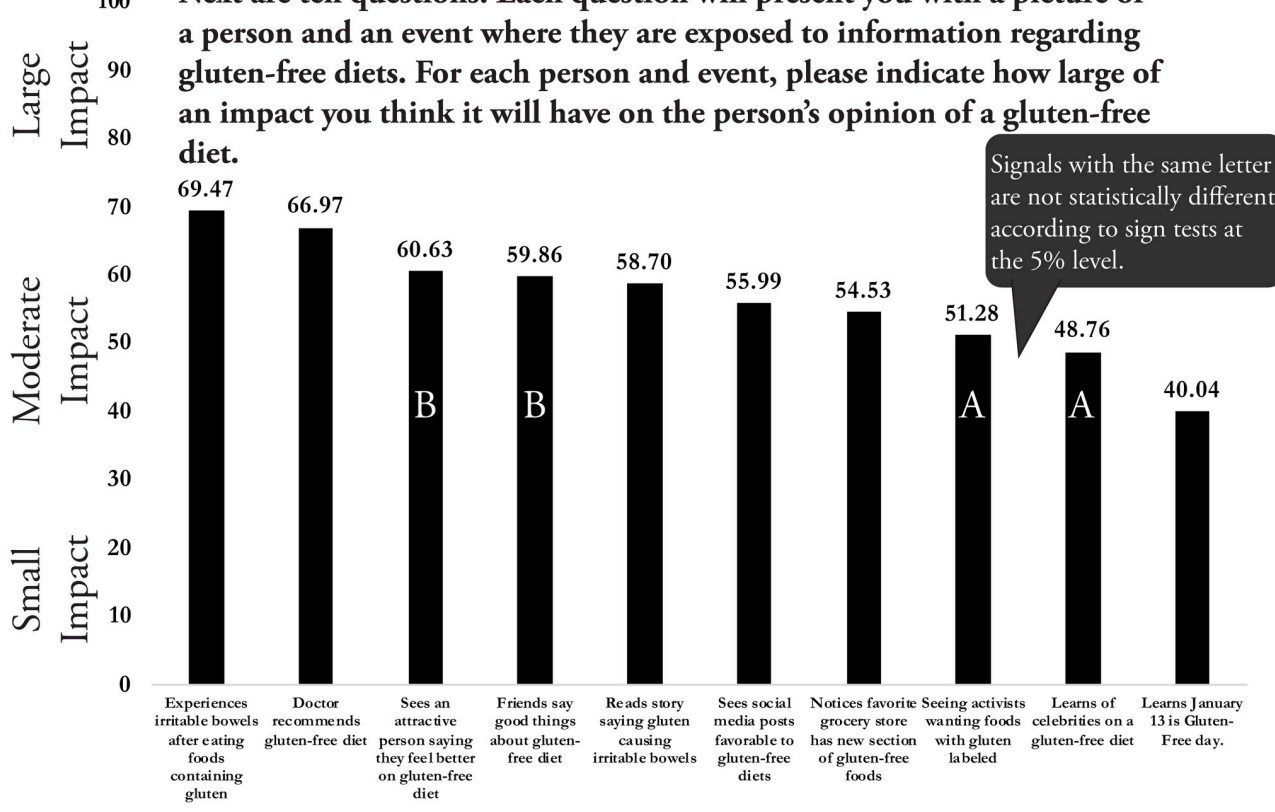

**Fig 3. Average impact score for information signals for 1,317 survey respondents.**

Each coefficient should be interpreted as the average impact score for that signal, assuming the default demographic. Each of these coefficients is roughly 4.37 units less than their respective average reported in Fig 3, so the overall results for the impact of the 10 signals are similar to the results in Fig 3.

The statistical significance of the signal coefficients does not provide meaningful information because it only says the impact scores are different from zero. What is more interesting is whether the coefficients for different signals are statistically different from each other. Wald tests are used to test the null hypothesis that the coefficient for any one signal is equal to the coefficient of another signal, and this test is performed for every combination of two signals. The null is rejected at the 5% level for every pair of signals except for *attractiverperson* and *friends*, so as with Fig 3, hearing good things about a gluten-free diet from an attractive person or one's friends has the same impact. All other signals have statistically different impacts.

The only statistically significant demographic variables are female and household size. From these coefficients we learn that female respondents tend to give slightly lower impact scores and households with three or more members tend to give slightly higher scores. The average differences for the impact scores between males and females, and between smaller and larger household sizes, are less than 8%, so the demographic effects are not remarkable. None of the variables referring to the hypothetical other persons are statistically significant, so while it was *a priori* deemed important to provide a visual description of the "other" person used in indirect questioning, *a posteriori* it does not appear necessary.

**Table 2. OLS regression using impact scores of information signals as dependent variable.**

| Variable | Coefficient estimate | Robust standard error |
|---|---|---|
| *personalexp* | 70.92** | 2.97 |
| *doctor* | 68.42** | 2.96 |
| *attractiveperson* | 62.08** | 2.95 |
| *friends* | 61.31** | 2.96 |
| *news* | 60.15** | 2.95 |
| *socialmedia* | 57.45** | 2.98 |
| *store* | 55.98** | 2.98 |
| *activist* | 52.73** | 2.97 |
| *celebrity* | 50.22** | 2.97 |
| *glutenfreeday* | 41.50** | 3.01 |
| *female* | -3.11** | 1.16 |
| *Less than 35 years of age* | -0.88 | 1.42 |
| *More than 54 years of age* | -2.49 | 1.43 |
| *White ethnicity* | -2.91 | 1.69 |
| *Black ethnicity* | 3.08 | 2.05 |
| *Hispanic* | 2.69 | 1.56 |
| *Household income > $75,000* | 2.67 | 1.41 |
| *Household income < $35,000* | 0.30 | 1.36 |
| *Northeast* | -0.27 | 1.79 |
| *Midwest* | 1.13 | 1.76 |
| *South* | 1.53 | 1.53 |
| *Household 3 or more members* | 3.17* | 1.28 |
| *Bachelors degree* | -0.50 | 1.21 |
| *person2* | 1.44 | 2.34 |
| *person3* | -0.91 | 2.53 |
| *person4* | 0.87 | 2.35 |
| *person5* | 0.17 | 2.57 |
| *person6* | 0.50 | 2.42 |
| *person7* | -2.69 | 2.65 |
| *person8* | -2.90 | 2.61 |
| *person9* | 1.19 | 2.29 |
| *person10* | -1.74 | 2.68 |
| *person11* | 0.69 | 2.55 |

N = 1,317 respondents and 13,170 observations.

* Denotes statistical significance at the 5% level.

** Denotes statistical significance at the 1% level.

## Regression 2: Results

The results of the weighted regression are shown below in Table 3. The second regression is similar to Regression 1 except that demographic variables are not used as demographic variables but are used instead for sample balancing. Because Regression 2 uses sample balancing weights, the regression coefficients should better reflect the beliefs of the average American as opposed to the beliefs of the average sample respondent. The interpretation of the coefficients is similar to that in Regression 1, except that the coefficients refer to the beliefs of the average American and not one particular demographic.

**Table 3. OLS weighted regression using impact scores of information signals as dependent variable.**

| Variable | Coefficient estimate | Robust standard error |
|---|---|---|
| *personalexp* | 71.92** | 2.01 |
| *doctor* | 69.70** | 2.01 |
| *attractiveperson* | 62.35** | 2.01 |
| *friends* | 62.21** | 2.01 |
| *news* | 60.65** | 2.00 |
| *socialmedia* | 58.78** | 2.03 |
| *store* | 57.21** | 2.05 |
| *activist* | 53.04** | 2.05 |
| *celebrity* | 51.11** | 2.06 |
| *glutenfreeday* | 41.77** | 2.13 |
| *person2* | 1.33 | 2.83 |
| *person3* | -4.06 | 3.00 |
| *person4* | -1.64 | 2.58 |
| *person5* | -4.32 | 3.25 |
| *person6* | -0.76 | 3.05 |
| *person7* | -4.58 | 3.37 |
| *person8* | -6.77* | 3.28 |
| *person9* | -0.61 | 2.72 |
| *person10* | -4.73 | 3.65 |
| *person11* | -0.07 | 3.08 |

N = 1,317 respondents and 13,170 observations.

* Denotes statistical significance at the 5% level.

** Denotes statistical significance at the 1% level.

While the coefficients for each signal differ in Regression 2 relative to Regression 1 (the average percentage difference is 13%), the overall results are similar. For example, the ranking of the signals with the highest and lowest scores are the same. The range of the signal coefficients is virtually identical as well. Wald tests (for the null hypothesis that the coefficients for any pair of signal coefficients are equal) are also identical to Regression 1, in that the only signals where the null is not rejected is for *attractiveperson* and *friends*. One difference from Regression 1 is that one of the *persons* variables is statistically significant. The coefficient for *Person8* (see Fig 2) has a lower value than the other persons, meaning that respondents felt this young man would be less influenced by any one signal than person. All the other *person* variables are not statistically different from zero.

## Sensitivity analysis: Results

Recall that 4.10% of the sample had never heard of gluten-free foods, so perhaps they should not be included in the analysis? To test the impact of including them, Regression 2 is estimated without these subjects, but this has virtually no impact on the results. The coefficients for any one signal changes by less than 1%, the ranking of the signal coefficients is unchanged, and the Wald tests for statistical differences between the signal coefficients are unchanged.

Another test for familiarity with gluten concerns subjects' abilities to pick the one food containing gluten from the list: bread, meat, honey, tomato, and lettuce. If Regression 2 is estimated using only those who correctly identified "bread" as having gluten, there are some changes in the magnitude of the coefficients for the 10 signals. Most change by less than 6%

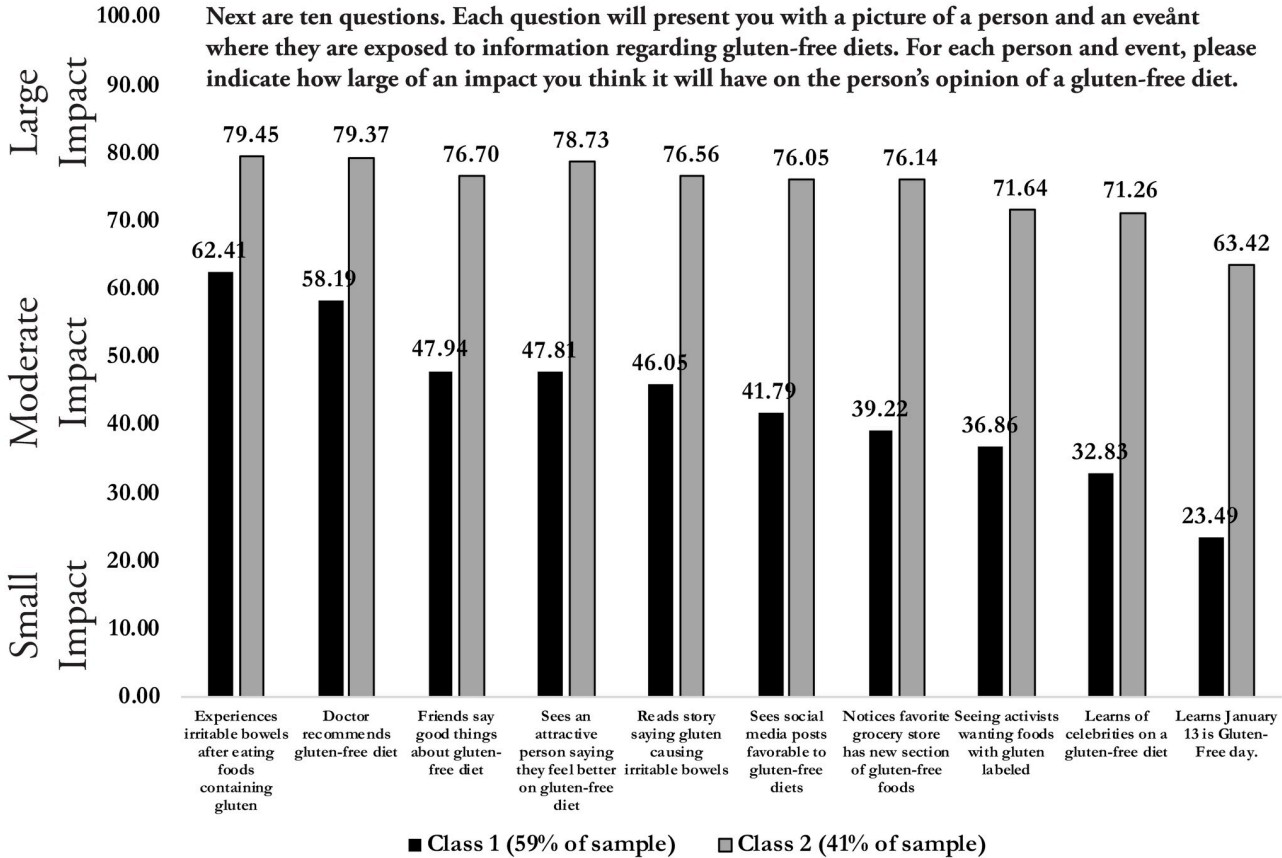

**Fig 4. Average impact score for information signals for 1,317 survey respondents, when respondents are separated into two classes with different survey response patterns.**

but one (*glutenfreeday*) increases 12%. However, the ranking of the most to least influential signals is unchanged, as are the Wald tests for statistical differences between signal coefficients.

Finally, a latent class model with two classes is estimated to help detect heterogeneity in beliefs about the impacts of the information signals. This model employs an OLS regression similar to Regressions 1 and 2 except (1) no sample balancing weights are used; (2) no explanatory variables are used other than the type of signal (no demographic variables or variables describing the hypothetical other); and (3) no robust standard errors to account for the panel nature in the data, but (4) two different sets of signal coefficients for two different classes of respondents who provide different patterns of responses. As such, the signal coefficients can be interpreted as the average impact scores for each class of respondents.

The model estimates suggest that, of all the respondents, 59% belong to Class 1 and 41% belong to Class 2. The OLS signal coefficients/average impact scores for each class are shown in Fig 4. The classes differ in two distinct ways. First, Class 1 provides lower impact scores for every signal, indicating they believe each of the 10 signals will have less of an impact on other people's opinions than Class 2. Second, Class 2 believes there will be greater heterogeneity of impacts across signals than Class 1. Except for *glutenfreeday*, Class 2 provides scores for all signals in the 70–80 range, whereas the range of scores for Class 1 is in the 30–60 range. Both classes agree that *glutenfreeday*, *celebrity*, and *activist* have the smallest impacts and *doctor* and *personal* have the largest impacts. From the latent class estimates we can conclude that the general results in the previous sections still hold true for almost half of the sample respondents.

For the other half, many of the results hold true, but there are fewer differences in the impact of the signals.

## Limitations

This study has a number of limitations which warrant acknowledgment. While the motivation for using indirect questioning has scientific justification, it would have been interesting to see how subjects would have responded if asked how they themselves would have responded to the information signals. Also, the impact of a signal was evaluated using only one type of question, whereas the results may have differed if other types were used. For example, this study allowed respondents to define a "large" or "small" impact however they liked, but different respondents may consider the same behavior to be different in terms of overall impact. Future research, including questions where a "large" impact, should be defined as "complete elimination of gluten from one's diet" and "small" impact as "avoiding eating gluten whenever it is convenient" would be useful in interpreting the results of the study and aid in validating the questions used.

## Discussion

What do these results tell us regarding the "truth" between gluten and health? We, of course, do not survey Americans in the search of an objective truth, but to better understand how consumers seek truth. Like scientists, most of our survey respondents would probably like to base their dieting behaviors on objective facts, or what philosophers call the Correspondence Theory of truth, whereby statements like "gluten is bad for you" is considered true if it corresponds to the actual state of affairs. This helps explain why respondents believe information signals like a doctor's recommendation or a personal eating experience would have the most impact on people's beliefs about the healthiness of gluten. Doctor recommendations are presumed to be based on scientific evidence, and while the feeling of an irritable bowel may technically be a subjective feeling, if a person feels irritable bowels that is what philosophers call first-person knowledge and there can be no ambiguity about what a person feels.

However, there is little scientific consensus on whether gluten sensitivity exists for non-Celiac patients, and no objective evidence that gluten is generally harmful to health for most people, so the decision to adopt a gluten-free diet cannot be based on the Correspondence Theory of truth. Instead, most people must rely on the Pragmatic Theory of truth, which contends "gluten is bad for you" is a true statement if it is useful—if it has value in some way to the believer [23]. There are many reasons why gluten may be deemed unhealthy in the pragmatic sense, and we explore some of these reasons below using the results of the survey.

It is sensible to take dieting advice from people similar to oneself, so the result that finds views on gluten are influenced by the eating habits of friends is intuitive. It is also sensible to take advice from people who seem to take good care of their bodies, which helps explain why equal importance is placed on the advice of attractive people and one's friends (and why commercials use attractive people as spokespersons). While social media does often spread misleading information, it can help people communicate valuable information as well, and the fact that social media posts touting a gluten-free diet are considered to have a larger influence than celebrities and activists is telling. All of these signals can be inferences that gluten can be harmful for health, and are based on the notion that other people have had personal experiences eliminating gluten from their diets, with positive results. Likewise, for a grocery store dedicating a new section exclusively to gluten-free diets—if it works for many other people, it might work for oneself.

It is important to recognize that the belief that gluten is bad for health may be useful even if it is not true. People will change their eating habits if they adopt this belief, and those new habits may be healthier even if gluten itself is healthy. The typical American diet is high in simple carbs, low in fiber, and lacking in diversity; this is partially due to America's high consumption of refined white flour as opposed to whole wheat flour. As one adopts a gluten-free diet, they can no longer consume wheat, rye, and barley products. So if they consume grains, they must find alternative sources such as buckwheat, teff, or quinoa. These alternative grains are more likely to be sold in whole form, containing the endosperm as well as the bran and germ. Thus, by eliminating gluten, one might be increasing one's consumption of fiber, vitamins, and other minerals—and increasing diet diversity at the same time.

So many food products contain gluten that it can be difficult to actually achieve a gluten-free diet unless one is very careful. Following a gluten-free diet involves being more intentional regarding one's food choices. This more serious attitude towards food and health is likely to lead to better food choices in general. It is also likely that, if one adopts a gluten-free diet in an effort to improve one's health, they may adopt other behavioral changes, like exercising more. If the adoption of a gluten-free diet occurs concomitant with these other changes, one's health may improve not because gluten was eliminated, but because of other reasons. Not recognizing this false association, though, people may attribute their health improvements—at least partially—to the absence of gluten. As they recommend a gluten-free diet to others, the demand for gluten-free products rise, grocery stores advertise more gluten-free products, people interpret this to mean gluten is harmful to health, so more people go gluten-free, and so on.

The point is that the belief that gluten is bad for health may be a useful belief even if it is not technically true. These beliefs are largely formed by interpreting social signals from specific people and the emergent behavior of social groups. Even seemingly insignificant signals like January 13 being the official gluten-free day is thought to have a moderate—not low—impact on beliefs about gluten. What this survey does is highlight the fact that a large number of respondents agree with the notion that beliefs about food and health stem not just from objective facts, but a variety of imperfect information signals emanating from society.

## Supporting information

**S1 Appendix.**
(DOCX)

## Author Contributions

**Conceptualization:** Franklin Bailey Norwood.

**Data curation:** Franklin Bailey Norwood.

**Formal analysis:** Franklin Bailey Norwood.

**Funding acquisition:** Franklin Bailey Norwood.

**Investigation:** Franklin Bailey Norwood.

**Methodology:** Franklin Bailey Norwood.

**Project administration:** Franklin Bailey Norwood.

**Resources:** Franklin Bailey Norwood.

**Software:** Franklin Bailey Norwood.

**Supervision:** Franklin Bailey Norwood.

**Validation:** Franklin Bailey Norwood.

**Visualization:** Franklin Bailey Norwood.

**Writing – original draft:** Franklin Bailey Norwood.

**Writing – review & editing:** Franklin Bailey Norwood.

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
