## [Decision Letter · Decision Letter 0]

2 Oct 2020

PONE-D-20-23496

Attractive people versus doctors: who knows the most about the impact of gluten on health?

PLOS ONE

Dear Dr. Norwood,

the two referees see merit, but also major issues that need to be improved. I would like to also stress the importance to properly and completely report methods and to further develop your result section. 

Thank you again for submitting your manuscript to PLOS ONE. We invite you to submit a revised version of the manuscript that addresses the points raised by each referee.

We look forward to receiving your revised manuscript.

Kind regards,

Christine Mohr, PhD

Academic Editor

PLOS ONE

Journal Requirements:

2. Please amend your current ethics statement to address the following concerns: Please explain why written consent was not obtained and if the ethics committees/IRBs approved this consent procedure.

3. Please address the following:

- Please include additional information regarding the survey or questionnaire used in the study and ensure that you have provided sufficient details that others could replicate the analyses. For instance, if you developed a questionnaire as part of this study and it is not under a copyright more restrictive than CC-BY, please include a copy, in both the original language and English, as Supporting Information. In addition, please include further details concerning the development and validation of this tool.

- Please ensure you have thoroughly discussed any potential limitations of this study within the Discussion section, including the potential non-naivety and trustworthiness of participants.

"Funding was made possible by the Barry Pollard MD / P&K Equipment Professorship in Agribusiness."

Additionally, because some of your funding information pertains to commercial funding, we ask you to provide an updated Competing Interests statement, declaring all sources of commercial funding.

In your Competing Interests statement, please confirm that your commercial funding does not alter your adherence to PLOS ONE Editorial policies and criteria by including the following statement: "This does not alter our adherence to PLOS ONE policies on sharing data and materials.” as detailed online in our guide for authors  http://journals.plos.org/plosone/s/competing-interests.  If this statement is not true and your adherence to PLOS policies on sharing data and materials is altered, please explain how.

Please include the updated Competing Interests Statement and Funding Statement in your cover letter. We will change the online submission form on your behalf.

Reviewers' comments:

Reviewer's Responses to Questions

**Comments to the Author**

1. Is the manuscript technically sound, and do the data support the conclusions?

Reviewer #1: Partly

Reviewer #2: Yes

2. Has the statistical analysis been performed appropriately and rigorously? 

Reviewer #1: No

Reviewer #2: Yes

3. Have the authors made all data underlying the findings in their manuscript fully available?

Reviewer #1: Yes

Reviewer #2: Yes

4. Is the manuscript presented in an intelligible fashion and written in standard English?

Reviewer #1: Yes

Reviewer #2: Yes

5. Review Comments to the Author

Reviewer #1: The topic tackled by this paper is an interesting one. The influences that have driven the trend around gluten-free diets (outside of diagnosed celiac disease) merits scrutiny. Though I think the value of this study is really in the potential application of this approach to the communication of food related information that has the potential to positively impact people’s diets. I do however have some major concerns about the methodology and the approach to data analysis. I think for this to be publishable standard these need to be revised and I would suggest a new data analysis strategy that better reflects the design of the study and the breadth of the dataset. Finally, I think it would be worth splitting the results and discussion sections and producing a more comprehensive discussion section.

1. The title feels a bit misleading at the moment. It needs to be made clearer that actual knowledge of attractive people/ doctors was not assessed – it was actually people’s perception of how impactful others would find their knowledge.

2. The introduction would benefit from a strong set of justified hypotheses in addition to the overarching aim stated.

3. Materials and methods -should be in past tense not present tense.

4. Please provide the exact number of respondents and the exact number of useable responses at line 73 rather than an ‘over 1500’ which is fine for a summary but doesn’t make sense in the context of a methods section.

5. Please provide the rationale for the sample size. Was a power analysis conducted?

6. Where did these questions that are referred to throughout the methods section come from? Were they made up specifically for this and how was this done? Was any kind of piloting or validation work engaged in?

7. Line 81: Did you check that they put people in a frame of mind thinking of gluten? If so, how? I would remove that claim otherwise.

8. What did you do with the 4% who had never heard of gluten free foods? Eliminate or retain and why?

9. How were the ten signals developed?

10. Was the scale (small to high impact) piloted or validated in any way?

11. Where did the pictures come from? How were they selected?

12. It would be really helpful to divide the methods section into ‘participants, materials, procedure and data analysis’ sections.

13. 93-105- this would sit better in the intro section – the overall justification of approach would give more of a justified rationale in the intro section and then the methods section could be more procedural. The nuts and bolts of what happened get a bit lost in the methods section at the moment.

14. There is currently no mention of ethics in the manuscript itself that I could see.

15. I would strongly recommend the results and discussion section be split into two separate sections. At the moment the combined section is quite confusing to follow and it also feels like the two functions of the results and the discussion are not currently adequately met.

16. If there were 11 person pictures and 10 signal statements, does that mean that across the sample not everyone saw every person? Doesn’t this mess up the counterbalancing and mean that you cannot really collapse across person for your analyses?

17. Overall, the approach to analysing the data is really disappointing. I really think a more sophisticated approach should be taken and I don’t really agree with this opinion that subjectivity around the scale is a reason to NOT engage in more sophisticated (probably parametric) analyses. It is a well anchored scale and it is not a between subjects design so differences in interpretation should average out? Notwithstanding this, subjectivity around scales is something that could be discussed in a limitations section of a discussion section.

18. Connected to the last point, it really seems odd not to analyse with ‘person’ as a factor in the analysis. I would strongly recommend that the author looks into more sophisticated ways of analysing data to better reflect the design of the study. Also, it seems a shame not to have taken account of other information collected such as demographics and attitudes of gluten free diets -are their individual differences for example? Can some specific hypotheses be developed in advance of scrutinising the data in this respect?

19. Please produce a more straightforward figure depicting key results. The background and extra bits make it difficult to scrutinise from a scientific perspective – attractive though it may look!

20. It doesn’t look like there was any engagement with open science- I would add this to a limitations section in the discussion section.

21. Overall, a discussion section which takes a more critical and less descriptive approach would be welcome. Also, more sophisticated analyses would allow for a deeper consideration of the overarching scientific questions.

22. Overall, a considered limitations section would be welcome.

23. Discussion of how this approach could be used on high priority questions around the communication of dietary information might be engaged in.

Reviewer #2: 1) Is the manuscript technically sound, and does the data support the conclusions?

In places yes, in other areas it seems very brief and could be improved.

There is no detail of survey method and indirect questioning in abstract.

The introduction gives a very simplistic overview of non coeliac gluten sensitivity and the controversy around this area - this lacks depth in my opinion and needs to be improved.

There's no mention of the criteria outlined for diagnosing NCGS see

Catassi et al. (2015) Diagnosis of Non-Celiac Gluten Sensitivity (NCGS): The Salerno Experts’ Criteria. Nutrients 7:4966-4977.

or of RCT that have found an effect from gluten - e.g.

Shahbazkhani et al (2015) Non-Celiac Gluten Sensitivity Has Narrowed the Spectrum of Irritable Bowel Syndrome: A Double-Blind Randomized Placebo-Controlled Trial. Nutrients (7) 4542-4554.

I am unclear if the survey had ethical approval and if participants gave consent

Or that Challenge amount used in gluten challenge is likely to be key re findings gluten challenge - This paper reports it's fructans rather than gluten but they use an amount for gluten challenge that is below what's recommended in the Catassi et al 2015 paper - so it is that surprising that they find no effect?

- Skodje GI, Sarna VK, Minelle IH, et al. Fructan, Rather Than Gluten, Induces Symptoms in Patients With Self-Reported Non-Celiac Gluten Sensitivity. Gastroenterology. 2018;154(3):529-539.e2. doi:10.1053/j.gastro.2017.10.040

Writing around weight loss @ line 49 is overly simplistic too. Excluded 'free from' diets are perceived by many to aid weight loss as they may restrict many foods that are treats and high in calories (ie. biscuits, cakes, sweets). Therefore there is evidence they are perceived as useful for weight loss.

You may find this paper a useful reference to motivations for following a gluten free diet in absence of coeliac disease.

Harper, L., Bold, J. (2018). An exploration into the motivation for gluten avoidance in the absence of coeliac disease. Gastroenterology and Hepatology from Bed to Bench 11 (3): 259-268.

More depth around responses would be useful too. E.g. Line 76-78 were responses representative geographically? Was this considered ? E.g. Rural v city areas, E / W coast versus central states?

6. PLOS authors have the option to publish the peer review history of their article (what does this mean?). If published, this will include your full peer review and any attached files.

Reviewer #1: **Yes: **Laura Wilkinson

Reviewer #2: No

---

## [Author Response · Author response to Decision Letter 0]

14 Dec 2020

I have attached a pdf document containing a letter to the editor and a Word document with my response to reviewers.

---

## [Decision Letter · Decision Letter 1]

2 Mar 2021

Perceived impact of information signals on opinions about gluten-free diets

PONE-D-20-23496R1

Dear Dr. Norwood,

We’re pleased to inform you that your manuscript has been judged scientifically suitable for publication and will be formally accepted for publication once it meets all outstanding technical requirements.

Kind regards,

Camelia Delcea

Academic Editor

PLOS ONE

Additional Editor Comments (optional):

Reviewers' comments:

Reviewer's Responses to Questions

**Comments to the Author**

1. If the authors have adequately addressed your comments raised in a previous round of review and you feel that this manuscript is now acceptable for publication, you may indicate that here to bypass the “Comments to the Author” section, enter your conflict of interest statement in the “Confidential to Editor” section, and submit your "Accept" recommendation.

Reviewer #1: All comments have been addressed

Reviewer #3: All comments have been addressed

2. Is the manuscript technically sound, and do the data support the conclusions?

Reviewer #1: Yes

Reviewer #3: Yes

3. Has the statistical analysis been performed appropriately and rigorously? 

Reviewer #1: Yes

Reviewer #3: Yes

4. Have the authors made all data underlying the findings in their manuscript fully available?

Reviewer #1: Yes

Reviewer #3: Yes

5. Is the manuscript presented in an intelligible fashion and written in standard English?

Reviewer #1: Yes

Reviewer #3: Yes

6. Review Comments to the Author

Reviewer #1: (No Response)

Reviewer #3: The article is very relevant to the current climate where people are easily influenced by diet trends. The article provides a quantitative perspective to this issue.

The revised version of the manuscript addresses the main issues that the previous revision brought up.

7. PLOS authors have the option to publish the peer review history of their article (what does this mean?). If published, this will include your full peer review and any attached files.

Reviewer #1: No

Reviewer #3: No

---

## [Editor Report · Acceptance letter]

29 Mar 2021

PONE-D-20-23496R1 

Perceived impact of information signals on opinions about gluten-free diets 

Dear Dr. Norwood:

I'm pleased to inform you that your manuscript has been deemed suitable for publication in PLOS ONE. Congratulations! Your manuscript is now with our production department. 

Kind regards, 

on behalf of

Dr. Camelia Delcea 

Academic Editor

PLOS ONE